# The Microbiome–Gut–Brain Axis and Dementia: A Bibliometric Analysis

**DOI:** 10.3390/ijerph192416549

**Published:** 2022-12-09

**Authors:** He-Li Sun, Yuan Feng, Qinge Zhang, Jia-Xin Li, Yue-Ying Wang, Zhaohui Su, Teris Cheung, Todd Jackson, Sha Sha, Yu-Tao Xiang

**Affiliations:** 1Unit of Psychiatry, Department of Public Health and Medicinal Administration, Institute of Translational Medicine, Faculty of Health Sciences, University of Macau, Macao SAR, China; 2Centre for Cognitive and Brain Sciences, University of Macau, Macao SAR, China; 3The National Clinical Research Center for Mental Disorders, Beijing Key Laboratory of Mental Disorders, Beijing Anding Hospital, Capital Medical University, Beijing 100054, China; 4Advanced Innovation Center for Human Brain Protection, Capital Medical University, Beijing 100054, China; 5School of Public Health, Institute for Human Rights, Southeast University, Nanjing 210096, China; 6School of Nursing, Hong Kong Polytechnic University, Hong Kong SAR, China; 7Department of Psychology, University of Macau, Macao SAR, China

**Keywords:** brain-gut axis, dementia, Alzheimer’s disease, bibliometrics, hotspots

## Abstract

Background: Associations between the microbiome–gut–brain axis and dementia have attracted considerable attention in research literature. This study examined the microbiome–gut–brain axis and dementia-related research from a bibliometric perspective. Methods: A search for original research and review articles on the microbiome–gut–brain axis and dementia was conducted in the Web of Science Core Collection (WOSCC) database. The R package “bibliometrix” was used to collect information on countries, institutions, authors, journals, and keywords. VOSviewer software was used to visualize the co-occurrence network of keywords. Results: Overall, 494 articles met the study inclusion criteria, with an average of 29.64 citations per article. Corresponding authors of published articles were mainly from China, the United States and Italy. Zhejiang University in China and Kyung Hee University in Korea were the most active institutions, while the *Journal of Alzheimer’s Disease* and *Nutrients* published the most articles in this field. Expected main search terms, “Parkinson disease” and “chain fatty-acids” were high-frequency keywords that indicate current and future research directions in this field. Conclusions: This bibliometric study helped researchers to identify the key topics and trends in the microbiome–gut–brain axis and dementia-related research. High-frequency keywords identified in this study reflect current trends and possible future directions in this field related to methodologies, mechanisms and populations of interest.

## 1. Introduction

A microbiome is a community of microorganisms, including bacteria, viruses, fungi, and protozoa, that forms in a specific environment [1]. In healthy human intestinal tracts, there are diverse and dynamic populations of microbiomes, most of which reside in the ileum and colon [2]. Previous studies have found that alterations of gut microbiota are associated with various intestinal disorders (e.g., irritable bowel syndrome) and brain function [3]. As such, the concept of “microbiome–gut–brain axis” has garnered growing attention and addresses complex bidirectional interactions between brain and gut [4]. For instance, the alteration or absence of an intestinal microbiome may trigger systemic immune activation that contributes to intestinal barrier defects, damage to the blood–brain barrier, the promotion of neuroinflammation, and eventual brain damage and degeneration [5].

Dementia is a chronic syndrome characterized by progressive deterioration of cognitive abilities [6]. Dementia can affect memory, thinking, language, behavior, and daily activities and is a major source of disability among older people worldwide [7]. Current estimates suggest that around 55 million people suffer from dementia globally and the number is projected to rise to 78 million by 2030 and 139 million by 2050 [8]. In addition, there are nearly 10 million new cases of dementia per year [8]. Alzheimer’s disease (AD) is the most common form of dementia and accounts for 60–80% of total cases [9]. Other forms of dementia include vascular dementia, Lewy body dementia, dementia in Parkinson’s disease, and frontotemporal dementia [10].

Some studies have implicated gut microbiota in the development or progression of dementia [11,12]. Longitudinal research has also shown that individuals with inflammatory bowel disease linked with gut microbiome were diagnosed with dementia at an earlier age on average compared with healthy samples [13]. Moreover, the microbiome–gut–brain axis has emerged as a potential diagnostic and therapeutic target in a wide range of psychiatric and neurologic disorders [4]. Relatedly, altering gut microbiota with antibiotic therapy or probiotic treatment can improve performance on learning and memory tests [14,15].

Understanding progress and trends within a particular research field is important. Bibliometric analysis is a widely used approach designed to identify key features of relevant publications including core research themes, methodologies, authors, institutions, and countries [16]. For instance, bibliometric analysis can provide information on citations of articles that reflect the academic impact of publications. In addition, the compilation of keyword frequencies can help researchers to identify past foci and future trends on specific research topics [17]. Bibliometric analysis also provides network maps of co-authorship and co-occurrence analysis that elucidate collaborations between countries, thus allowing researchers to seek potential interdisciplinary collaborators [18].

Previous bibliometric manuscripts have explored associations between gut microbiota and Parkinson’s disease [19], associations between the gut–brain axis and depression [20], and the microbiome–gut–brain axis [21]. However, to date, links between the microbiome–gut–brain axis and dementia have not been analyzed using the bibliometric method. Therefore, this study examined characteristics of relevant research on the microbiome–gut–brain axis and dementia from a bibliometric perspective.

## 2. Materials and Methods

### 2.1. Data Collection

Relevant studies on the microbiome–gut–brain axis and dementia were extracted from the Web of Science Core Collection (WOSCC) database; the editions were limited to Science Citation Index Expanded (SCI-EXPANDED) and Social Sciences Citation Index (SSCI), which are the most widely used sources for bibliometric analyses. Following previous studies [21,22], the key terms “microbiome–gut–brain axis” and “dementia” and its synonyms were used as search terms, with the search strategy: TS = (microbiome–gut–brain axis OR axis–brain–gut OR microbiota–gut–brain) AND TS = (dementia OR Alzheimer OR Alzheimer’s disease OR AD). Original articles and reviews published in English were searched from dates of inception to 30 July 2022. Bibliometric data were exported as “plain text” documents with the full record and cited references. Overall, 494 publications on microbiome–gut–brain axis and dementia were included for analyses; 245 were original articles and 249 were reviews.

### 2.2. Data Analysis

Bibliometrix is an R-based tool for comprehensive bibliometric mapping analysis [23] that can be used for (1) number of publications, (2) citation analysis including summaries of citation information for countries or regions, affiliations, journals, and authors, and (3) the conduct of Three-Fields Plots from the keywords Plus analysis. VOSviewer is a software program based on the Java environment [24] used to (1) create clusters of keywords and (2) perform co-occurrence analyses of keywords. We also used an online bibliometric analysis approach (https://bibliometric.com/, accessed on 30 July 2022) to generate a string plot of research collaborations between countries. Microsoft Excel 2019 was used to map the annual number of publications and depict a circle plot of country contributions and bar plot of keyword frequencies. Moreover, we used a knowledge synthesis method to summarize specific keywords used in this literature [25]. Figure 1 presents a flowchart of publication selection and analysis.

## 3. Results

### 3.1. General Features of Publications

Overall, 494 publications on microbiome–gut–brain axis and dementia-related research were published between 2002 and 2022, with an annual growth rate of 25.6%. These publications included 245 (49.6%) original articles and 249 (50.4%) reviews involving 246 sources (Journals and Books), 2986 authors, and 34,963 references; the average number of citations for each publication was 29.64. As shown in Figure 2, the publication total increased substantially, particularly in recent years, from 20 publications in 2017 to 145 in 2021.

### 3.2. Top Cited Papers

Table 1 presents the 10 most cited articles in the field of microbiome–gut–brain axis and dementia-related research, each of which had total citation counts (TC) of at least 200. The two most highly cited articles were general review papers on the microbiome-gut brain axis. First, was a review entitled, “The Microbiota-Gut-Brain Axis” [26] that was published in *Physiological Reviews* and generated 894 citations. Second, was a review, “Interactions between the microbiota, immune and nervous systems in health and disease”, published in Nature Neuroscience, that garnered 764 citations. The 8 remaining most highly cited papers were all review papers as well but focused more specifically on links between the microbiome–gut–brain axis and Alzheimer’s disease or neurogenerative diseases; each had generated roughly 200–330 citations to date. In terms of total citations per year (TC/Y), “The Microbiota-Gut-Brain Axis” also ranked first with a TC/Y of 223.5, followed by “Interactions between the microbiota, immune and nervous systems in health and disease” (TC/Y = 127.33) and “Brain-Gut-Microbiota Axis in Alzheimer’s Disease” (TC/Y = 69.25) [27] published in the *Journal of Neurogastroenterology and Motility*.

### 3.3. Journal Analysis

Table 2 presents the 15 most cited journals out of 246 journals on microbiome–gut–brain axis and dementia-related research. The *Journal of Alzheimer’s Disease* was ranked first in both total citations (TC = 620) and number of publications (N = 23), followed by *Nutrients* (TC = 423, N = 19). However, *Nutrients and Frontiers in Aging Neuroscience* had the highest H-index (9) compared with other journals; over half of the 15 most cited journals had H-indexes ranging between 2 and 4. With respect to impact factors (IF), *Frontiers in Immunology* ranked highest (IF = 8.786), followed by *Antioxidants* (IF = 7.765) and *Nutrients* (IF = 6.706). Of the 15 most highly cited journals, 5 are in the category of Medicine, 4 are in Agricultural and Biological Sciences, and two are in the field of Neuroscience. All journals are located in quartiles 1 or 2 according of their respective Journal Citation Reports categories (2021–2022), except for *Microorganisms*, a Q3 journal with an IF of 4.926.

### 3.4. Author Analysis

Table 3 summarizes the 10 most influential authors in the field of microbiome–gut–brain axis and dementia-related research. Dr. Katerina Johnson from Oxford University had the highest total citation (TC = 552) and average total citation (ATC = 138) counts, followed by Dr. Sunmin Park from Hoseo University (TC = 178, ATC = 35.6). There is a marked difference between Dr. Johnson’s ATC and that of other top authors. Of the top 10 authors, nine had published 5 papers while Dr. Johnson had a much higher number of citations from just 4 publications. In terms of H-index rates, there was little difference between the 10 authors; 6 authors had an H-index of 5, 3 authors had an H-index of 4 and one author had an H-index of 3.

### 3.5. Country/Territory Analysis

Figure 3 shows country/territory contributions to microbiome–gut–brain axis and dementia-related research. A total of 46 countries/territories had published at least one paper in this field. Figure 3 suggests that China had the highest number of publications (NP) with 136 articles, followed by the United States (U.S.) (92 articles), and Italy (46 articles). Countries with more than 10 publications included Korea, Spain, Germany, Poland, the United Kingdom, Australia, and Canada. Figure 3 shows the top 15 most cited countries/territories. Of the 46 countries/territories with publications on microbiome–gut–brain axis and dementia, four had no citations to date. Of the top 10 most cited countries/territories with at least 340 total citations (TC), four had 1000 citations or more, including the U.S. (TC = 3301), China (TC = 2767), Italy (TC = 1857), and Ireland (TC = 1036). Regarding differences between the ranking of average article citations (AAC) and TC, Ireland showed the largest gap with 148 AAC and a rank of fourth in TC. Austria was ranked second in AAC (117.33), but ninth on the TC rankings. Inconsistencies between AAC and TC may be due to differences in high quality research articles between these countries/territories. Figure 4 shows the international collaborations in microbiome–gut–brain axis and dementia related research. China and the U.S. had the most extensive cooperation in microbiome–gut–brain axis and dementia research, with 23 publications. U.S. institutions also worked closely with those in Germany (N = 9), UK (N = 8), Canada (N = 7), and Brazil (N = 6).

### 3.6. Institution Analysis

Figure 5 illustrates institution contributions to microbiome–gut–brain axis and dementia-related research. In total, 924 institutions contributed to this research area. The right and left panels, respectively, show the number of publications by institution and annual publications for each institution. Zhejiang University was ranked first and showed a rapid rate of growth in annual publications from two papers in 2014 to 26 articles in 2022. Kyung Hee University was ranked second, followed by University of College Cork. The top nine most active institutions began publishing related articles after 2014, with Duke University and the University of Texas Health Science Center, Houston both publishing their first related articles in 2018.

### 3.7. Keyword Analysis

Figure 6 shows the most frequently used keywords in the field. Of 1116 unique keywords, “gut microbiota” and “Alzheimer’s disease” were used most frequently, with 114 and 108 occurrences, respectively, followed by “gut-brain axis” with 97 occurrences. “Parkinson’s disease” as a common neurodegenerative disease had 81 occurrences. “Chain fatty acids” and “Amyloids” as major features of AD pathology had 72 and 44 occurrences, respectively. Several keywords related to hypothesized dementia mechanisms, including “inflammation” and “oxidative stress”, were also used as frequent keywords, with 64 and 49 occurrences, respectively. The middle and bottom panels of Figure 6 present the co-occurrence analysis of keywords, with a minimum number of 8 occurrences. After keywords with similar meanings were merged, 37 keywords were retained for analyses. The size of nodes in Figure 6 represents the frequency of keywords while the color of nodes represents the correlation between nodes. All keywords could be divided into 3 main clusters. The red cluster was the largest one and included several neurodegenerative diseases such as “Alzheimer’s disease”, “Parkinson disease”, “multiple sclerosis” as well as inflammation related terms such as “inflammation” and “oxidative stress”. The green cluster included two axes (“gut–brain axis” and “microbiota–gut- brain axis”) and two systems (“immune system” and “central nervous system”). The blue cluster included “mild cognitive impairment” and “cognition”, both of which were related to cognitive function.

## 4. Discussion

This is the first bibliometric analysis of research on the microbiome–gut–brain axis and dementia. Of 494 publications identified from the WOSCC database search, around half were original research articles and the other half were review articles. The results showed a substantial growth of relevant publications, particularly during the period from 2017 to 2021.

The total number of citations can be used as an important indicator of interest in a particular research area [28]. The top 10 most cited articles in this study were all review papers that emphasized the importance of the microbiota–gut–brain axis and/or its links with neurodegenerative disease development. Sharp increases in the number of papers and citations over the past several years provide evidence of the rapid growth in interest in this field. The brain–gut axis reflects bidirectional communication between central and enteric nervous systems [26]; as one of the main regulators of the brain–gut axis, the distribution of neuroactive compounds released by microbiota around the axis may lead to the changes of cognitive function that contribute to the development of dementia [29]. Specifically, dementia is hypothesized to arise when gut bacteria activate immune activation through a defective intestinal barrier and lead to systemic inflammation, that, in turn, disrupts the blood–brain barrier and promotes neuroinflammation, ultimately leading to nerve damage and degeneration [30,31,32]. Evidence based on a nationwide population-based cohort study supports this hypothesis, as irritable bowel syndrome patients are at higher risk of dementia compared to healthy controls [33].

Regarding publication outlets for microbiome–gut–brain axis and dementia research, different dimensions were explored to identify journal contributions to the field. The H-index is an indicator of research performance, incorporating quantity and impact of publications into a ratio [34]. For the journal *Nutrients*, the H-index of 9 indicates 9 papers, each having at least 9 citations, having been published to date. Regarding the IF value and JCR classification of journals, higher IF values do not always represent higher JCR classifications; for example, the Q2 journal *Aging-US* has a higher IF than the Q1 journal *Current Alzheimer Research*, because the journals are included in different research domains. Notably, compared with other bibliometric analyses of journals typically found in a single research domain [35], research on the microbiome–gut–brain axis and dementia is not limited to medicine and extends to other fields including Agricultural and Biological Sciences, Nutrition and Dietetics, and Neuroscience. This pattern could reflect relatively high levels of interdisciplinary focus in the field. As such, researchers pursuing this field may need to be literate in the content and typical methodologies of different fields.

China is currently the most productive country in the number of publications on microbiome–gut–brain axis and dementia research, followed by the U.S. Generally, the economic situation of a country affects the level and priorities in research funding support, thus influencing its academic capability [36]. For instance, the National Institute of Health in the U.S. launched a strategically focused research program called the Human Microbiome Project in 2007; the program has provided financial support and attracted scholars in microbiome related fields [37] as a result. Many countries have joined this project and developed collaborations with U.S.-based institutions, especially China and Germany. Microbiome–gut–brain axis research had become more popular globally as a result of such initiatives [38,39].

Keywords can be regarded as core content of a specific article. Hence, keyword frequencies provide important clues about major trends in a given research area [40]. Apart from “gut microbiota”, “Alzheimer’s disease” was the most frequently used keyword in the field of microbiome–gut–brain axis and dementia. As the most common form of dementia, AD has far-reaching implications for brain health and function in the overall population [41]. A core feature of AD is the severe accumulation of senile plaques involving amyloid β protein (Aβ) and neurofibrillary tangles [42]. Notably, “amyloids” were another frequently used keyword in this research field and reflect a trend in research linking changes in gut microbiota strains with the progress of amyloid deposition [43]. Compared with healthy controls, AD patients display less microbial diversity of gut microbiota including deficits in fusobacteriaceae, firmicutes, and actinobacteria [44,45,46]. However, mechanisms underlying connections between gut microbiome and AD are still not clear. Some authors assume that inflammation plays a vital role [46,47]. The intestinal gut is involved in regulating the development and function of microglia. Presumably, its repeated activation has a wide range of proinflammatory effects that may lead to the progression of AD [48]. In addition, “Chain fatty acids” was another high frequency keyword; short chain fatty acids (SCFA) are key gut microbiota metabolites that typically grow in the large intestine [49]. Previous studies have found that SCFAs play an important role in the microbiota–gut–brain axis and have been linked with neurodegenerative diseases including AD [50]. One metabolomics study found that the concentration of propionate (a type of SCFA) was higher in the saliva of AD patients than healthy controls [51]. Similarly, compared with healthy controls, acetic acids level may be elevated in AD patients [52].

Intestinal gut dysbiosis is also associated with various diseases including irritable bowel syndrome (IBS). As another commonly used keyword in microbiome–gut–brain axis and dementia research, “oxidative stress” has been proposed as a potentially important risk factor for IBS in the field [53]. Over time several studies documented the co-occurrence of IBS and AD [13,54,55]. From this foundation, a recent genome-wide cross-trait analysis indicated AD and gastrointestinal tract disorders share several loci including PDE4B and BRINP3 that are promising candidates as underlying gut–brain mechanisms related to AD and IBD [56]. As another high frequency keyword, “Parkinson’s disease” (PD) is a progressive neurodegenerative disorder displaying an altered α-synucleinopathy that may be distributed along the brain–gut axis [57]. Dysregulation of the microbiota–brain–gut axis is common among PD patients [58] and is a concomitant symptom for over 80% of these cohorts [59]. A meta-analysis involving more than 30,000 participants revealed implicated helicobacter pylori infection as an agent associated with increased PD risk [60]. Furthermore, a randomized control trial found that eradicating Helicobacter pylori with L-dopa can improve PD symptoms [61].

Another trend identified in the keyword analysis was “mouse model”. Animal models have been widely used to examine associations of the gut microbiome with dementia [62]. One of the most widely used mouse models in this field is the APP/PS1 double transgenic mice, an AD-like model [63]. In this type of mouse, the presence of mutated transgenes can induce Aβ plaque deposition, resulting in memory deficits similar to those observed among humans with AD [64]. Alternately, a murine AD model posits that gut microbiome composition influences the build-up of amyloid plaque deposition, and oral antibiotic treatments might reduce such progress [47]. Still other animal studies have found that prebiotics, such as Lactobacillus strains, have benefits for brain function through the gut–heart–brain axis and can be used as a therapeutic agent for cognitive impairment [65].

Notwithstanding its implications for research, this study has several limitations. First, considering bibliometric analysis recommendations [66] that using a single database is preferable [67], data were only searched and extracted exclusively from the most widely used source, the WOSCC database. Second, bibliometric methods cannot provide a nuanced summary of salient subject content based on a large number of studies, as sometimes the document type defined by the WOSCC may be inaccurate [21].

## 5. Conclusions

In conclusion, this bibliometric analysis of microbiome–gut–brain axis and dementia has shown a rapid increase in the number of associated publications during the past 10 years; in particular, there has been a sharp rise in papers published during the two most recent years that underscores the current popularity of this field. Zhejiang University in China and Kyung Hee University in Korea were the most active institutions, while the *Journal of Alzheimer’s Disease* and *Nutrients* were the most active journals in microbiome–gut–brain axis and dementia research. Keyword analysis results revealed “Alzheimer’s Disease”, “Parkinson disease”, “chain fatty-acids”, “inflammation”, and “mouse model” as high-frequency keywords that reflect current trends and possible future directions in this field related to populations of interest, mechanisms, and methodologies.

## Figures and Tables

**Figure 1 ijerph-19-16549-f001:**
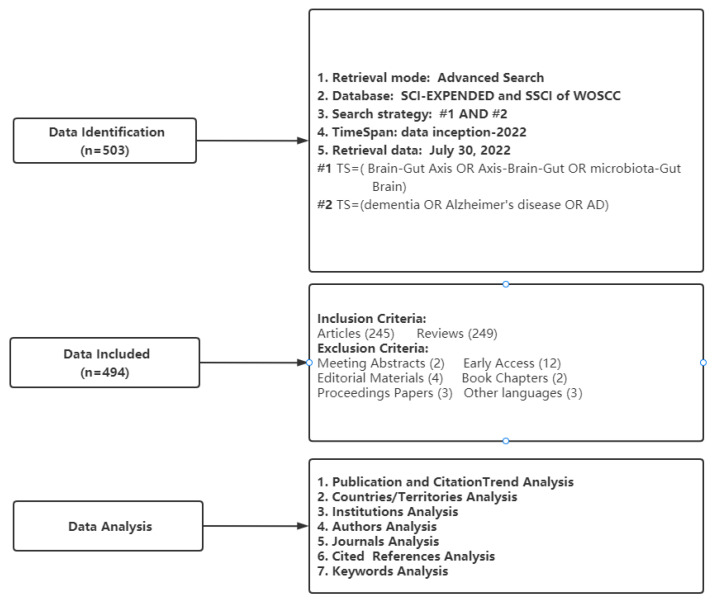
Flowchart of data selection and analysis.

**Figure 2 ijerph-19-16549-f002:**
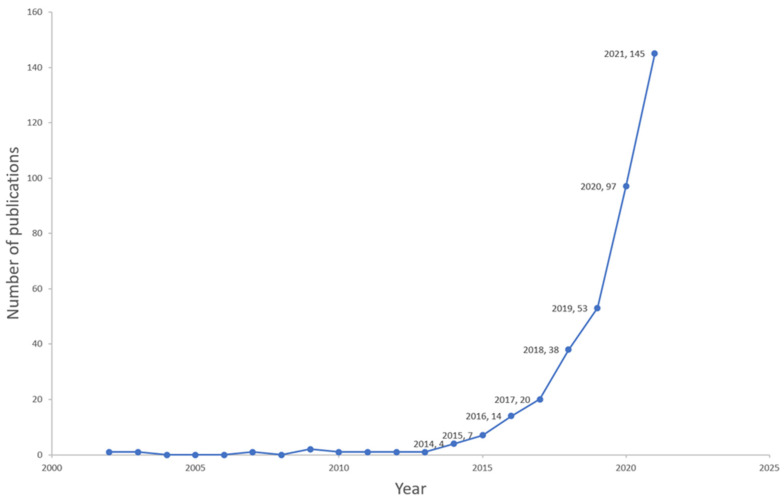
Year-wise distribution of the documents between 2002–2021.

**Figure 3 ijerph-19-16549-f003:**
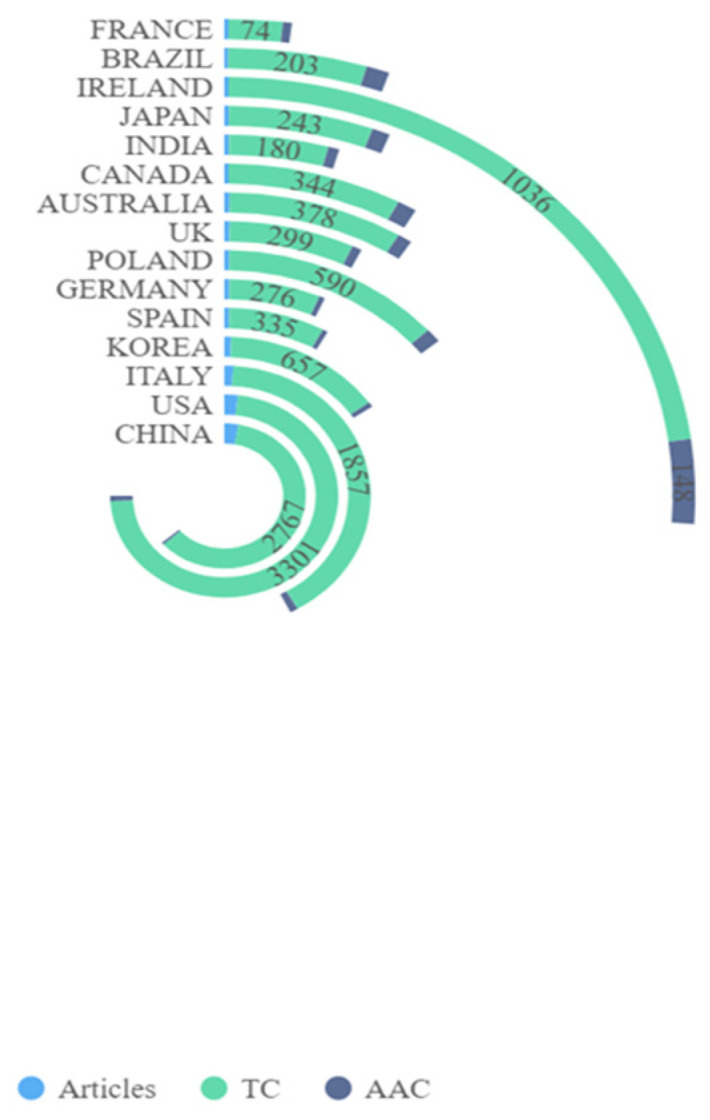
Top 15 contributing countries on microbiome–gut–brain axis and dementia-related research. Notes: TC: Total number of citations; AAC: Average Article Citations.

**Figure 4 ijerph-19-16549-f004:**
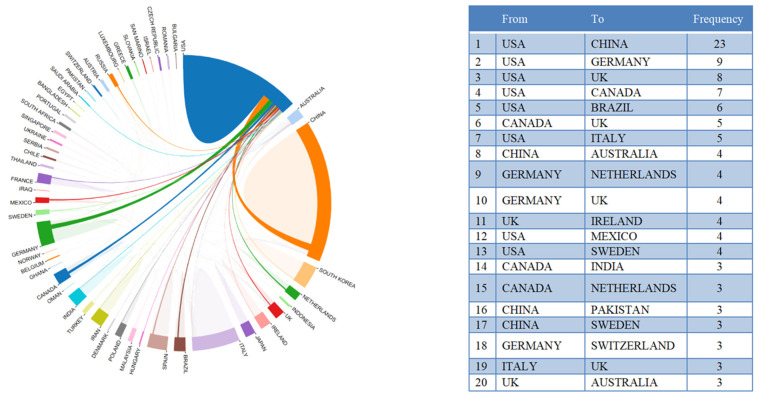
International collaboration analysis and the top 20 collaborations between countries.

**Figure 5 ijerph-19-16549-f005:**
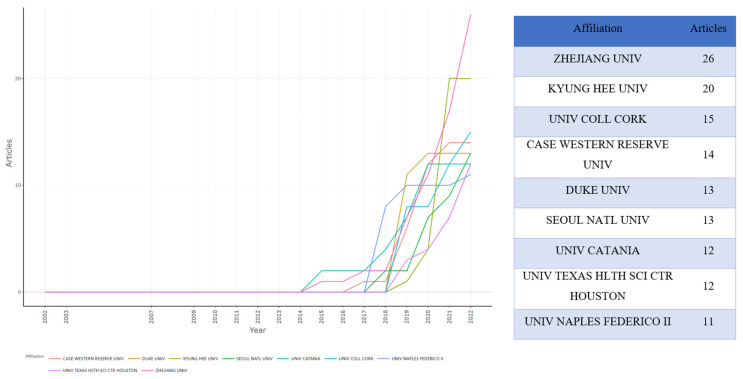
Top 9 contributing institutions and production over time on microbiome–gut–brain axis and dementia-related research.

**Figure 6 ijerph-19-16549-f006:**
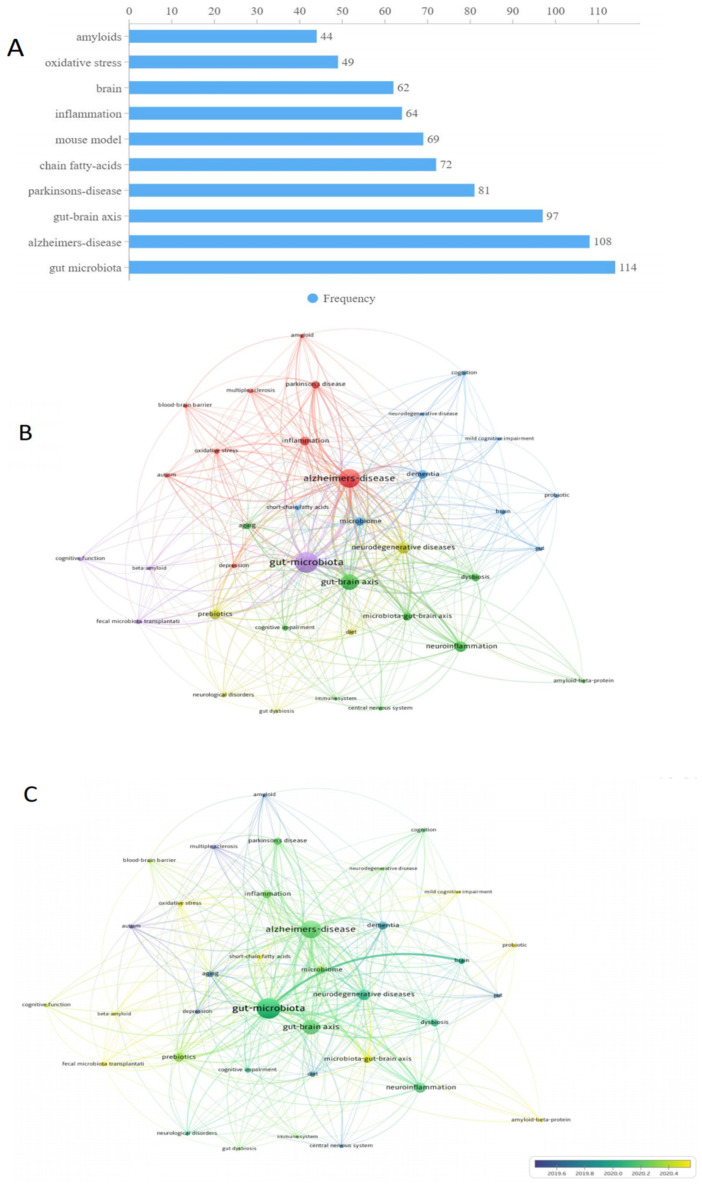
Co-occurrence analysis of author keywords and top 10 keywords on COVID-19 and sleep related research. ((**A**): the most frequently used keywords. (**B**): Co-occurrence network analysis of author keywords. (**C**): Co-occurrence overlay analysis of author keywords).

**Table 1 ijerph-19-16549-t001:** Top 10 most cited articles on microbiome–gut–brain axis and dementia related research.

SCR	Title (N = 494)	Year	Journal	FA	CA	TC	TC/Y
1	The Microbiota-Gut-Brain Axis	2019	*Psychological Review*	John F. Cryan	Timothy G. Dinan	894	223.50
2	Interactions between the microbiota, immune and nervous systems in health and disease	2017	*Nature Neuroscience*	Thomas C Fung	Elaine Y Hsiao	764	127.33
3	The Gut Microbiota and Alzheimer’s Disease	2017	*Journal of Alzheimers Disease*	Chunmei Jiang	Bin Zhao	329	54.83
4	Brain-Gut-Microbiota Axis in Alzheimer’s Disease	2019	*Journal of Neurogastroenterology and Motility*	Karol Kowalski	Agata Mulak	277	69.25
5	Microbiota-Microbiome-Gut-Brain Axis and Neurodegenerative Diseases	2017	*Current Neurology and Neuroscience Reports*	Eamonn M M Quigley	Eamonn M M Quigley	263	43.83
6	Gut microbiome in health and disease: Linking the microbiome-gut-brain axis and environmental factors in the pathogenesis of systemic and neurodegenerative diseases	2016	*Pharmacology* *& Therapeutics*	Shivani Ghaisas	Anumantha Kanthasamy	252	36.00
7	Role of gut microbiota and nutrients in amyloid formation and pathogenesis of Alzheimer disease	2016	*Nutrition Review*	Francesca Pistollato	Maurizio Battino	250	35.71
8	The bowel and beyond: the enteric nervous system in neurological disorders	2016	*Psychiatry Research*	Meenakshi Rao	Michael D Gershon	230	32.86
9	Neuropeptide Y: A stressful review	2016	*Neuropeptides*	Florian Reichmann	Peter Holzer	227	32.43
10	Microbiome, probiotics and neurodegenerative diseases: deciphering the gut brain axis	2017	*Cellular and Molecular Life Sciences*	Susan Westfall	Satya Prakash	211	35.17

Notes: SCR: Standard competition ranking; TC: Total Citations; TC/Y: Average per Year Total Citations; FA: First author; CA: Corresponding author.

**Table 2 ijerph-19-16549-t002:** Top 15 most cited journal contributed to microbiome–gut–brain axis and dementia related research.

SCR	Journal (N = 246)	h_index	TC	NP	IF	JCR	Research Domain
1	*Journal of Alzheimers Disease*	8	620	23	4.160	Q1	Psychology
2	*Nutrients*	9	423	19	6.706	Q1	Agricultural and Biological Sciences
3	*International Journal of Molecular Sciences*	7	237	18	6.208	Q1	Chemistry
4	*Frontiers in Aging Neuroscience*	9	386	14	5.702	Q1	Neuroscience
5	*Frontiers in Neuroscience*	5	103	12	5.152	Q2	Neuroscience
6	*Microorganisms*	6	83	8	4.926	Q3	Medicine
7	*Antioxidants*	4	45	7	7.675	Q1	Agricultural and Biological Sciences
8	*Current Alzheimer Research*	4	155	7	3.040	Q1	Medicine
9	*Frontiers in Immunology*	3	82	7	8.786	Q1	Medicine
10	*Aging-Us*	4	159	6	5.955	Q2	Biochemistry, Genetics and Molecular Biology
11	*Frontiers in Cellular and Infection Microbiology*	4	164	6	6.073	Q2	Medicine
12	*Frontiers in Nutrition*	4	32	6	6.590	Q1	Food Science
13	*Frontiers in Pharmacology*	2	49	6	5.988	Q1	Medicine
14	*Journal of Agricultural and Food Chemistry*	3	62	6	5.895	Q1	Agricultural and Biological Sciences
15	*Scientific Reports*	4	385	6	4.996	Q1	Multidisciplinary

Notes: SCR: Standard competition ranking; TC: Total citations; NP: Number of publications; IF: Impact factor (2021–2022); JCR: Journal Citation Reports category (2021–2022).

**Table 3 ijerph-19-16549-t003:** Top 10 authors contributed to microbiome–gut–brain axis and dementia related research.

Name	Institution	h_index	NP	TC	ATC
Johnson K Katerina V.-A. Johnson	University of Oxford	4	4	552	138
Park S Sunmin Park	Hoseo University	4	5	178	35.6
Goebel Stengel M Miriam Goebel-Stengel	Martin-Luther-Krankenhaus	5	5	105	21.0
Kobelt P Peter Kobelt	Charité—Universitätsmedizin Berlin	5	5	105	21.0
Rose M	Philipps-Universität	5	5	105	21.0
Stengel A Andreas Stengel	University Hospital Tübingen	5	5	105	21.0
Liu X Xiaofei Liu	South China University of Technology	4	5	96	19.2
Perry G George Perry	The University of Texas at San Antonio	5	5	89	17.8
Obrenovich M Mark Obrenovich	Cleveland State University	5	5	82	16.4
Zhang M Meng Zhang	Beijing Gene Tangram Technology	3	5	78	15.6

Notes: NP: Number of publications; TC: Total citations; ATC: Average Total citations.

## Data Availability

Not applicable.

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
