# Peer review of "The Microbiome–Gut–Brain Axis and Dementia: A Bibliometric Analysis"

_ijerph, 2022, doi:10.3390/ijerph192416549_

Round 1
Reviewer 1 Report
Line 22: Please change "regular papers" to "original research"
Figure 1:
Under "Data Identification" (n=503): From where was this number found? Please make it clear in the text.
Please mention the inclusion and exclusion criteria in the text along with the number of records obtained.
Under "Data Analysis": Please check the spelling of Publication and correct it.
Line 138: Please check the reference style (Fung et al., 2017) as per the journal.
Table 1: Year of publication may be added to this list.
Line 152: "629 journals": This journal number does not match with the total number of sources as mentioned in Line no: 125. Please check and correct it.
Table 3: Please arrange the name of authors either as per h-index or ATC.
Line 260-263: Authors have mentioned the top 10 most cited articles in the area, which are obviously "review papers". My request here is to discuss a brief about the present status of original research in this area along with future directions in this aspect under this section.
Author Response
Response to Reviewer 1 Comments
We’d appreciate sincerely your effusive work and invaluable comments for our research. We took seriously your comments and criticism and revised the manuscript in response to your suggestions. We have revised the manuscript according to the remarks and highlighted the changes in yellow color for easy inspection. A detailed point-by-point response can be seen below.
- Line 22: Please change "regular papers" to "original research"
Response 1: thank for your suggestion, we have changed in the final manuscript.
- Figure 1: Under "Data Identification" (n=503): From where was this number found? Please make it clear in the text.
Response 2: As recommended, the details of the data identification for literature search have been mentioned in “Date collection” part. (Page 2)
- Please mention the inclusion and exclusion criteria in the text along with the number of records obtained.
Response 3:
- Under "Data Analysis": Please check the spelling of Publication and correct it.
Response 4: thank for your reminder, we have revised it to number of publication. (Page 3)
- Line 138: Please check the reference style (Fung et al., 2017) as per the journal.
Response 5: we have revised it. (Page 3)
- Table 1: Year of publication may be added to this list.
Response 6:Following the recommendation, we have added year of publication in Table 1. (Page 5)
- Line 152: "629 journals": This journal number does not match with the total number of sources as mentioned in Line no: 125. Please check and correct it.
Response 7:It is a typo error and we have revised the to “246” in the main text. (Page5)
- Table 3: Please arrange the name of authors either as per h-index or ATC.
Response 8: we have rearranged the authors name based on ATC.
- Line 260-263: Authors have mentioned the top 10 most cited articles in the area, which are obviously "review papers". My request here is to discuss a brief about the present status of original research in this area along with future directions in this aspect under this section.
Response 9: As recommended, we have added the present status of original research in the text. (Page 3)
Reviewer 2 Report
This article is well written, and gives a novel perspective to explore the relationship of microbiome-gut-brain axis and dementia. However, this paper does not highlight the significance of research, and there are some minor issues should be improved.
Major issue
1 I don't think it is very meaningful to list the authors and magazines most related to the research of dementia and gut-brain-axis. But it is instructive to find some key words, such as "chain fatty acids" and “Amyloids”, which have guiding significance on the future research about microbiome-gut-brain axis and dementia. Therefore, I hope the author will give more discussion on the part of “Keywords”, and it is better to be more detailed to highlight the significance of this study.
Minor issues
1 There are spelling errors in Figure 1.
2 The words in Figures 5 and 6 are not clear enough, and the font is too small to see clearly.
Author Response
Reply to the Reviewer 2
- I don't think it is very meaningful to list the authors and magazines most related to the research of dementia and gut-brain-axis. But it is instructive to find some key words, such as "chain fatty acids" and “Amyloids”, which have guiding significance on the future research about microbiome-gut-brain axis and dementia. Therefore, I hope the author will give more discussion on the part of “Keywords”, and it is better to be more detailed to highlight the significance of this study.
Response 1: thank for your suggestion, we have discuss all the top 10 keywords.(Page 12)
- There are spelling errors in Figure 1.
Response 2: we have revised the spelling error.
- The words in Figures 5 and 6 are not clear enough, and the font is too small to see clearly.
Response 3: we have uploaded the original figure. (Page 10)